# The Field of Gender Through Metaphors: The Dilemma of Female and Male Referees in the Minds of Football Fans

**DOI:** 10.3390/bs15101359

**Published:** 2025-10-05

**Authors:** Sabiha Gizem Engin

**Affiliations:** Faculty of Sports Sciences, Yozgat Bozok University, Yozgat 66100, Türkiye; s.gizem.engin@bozok.edu.tr

**Keywords:** football, gender norms, referee, sexual double standards, metaphor analysis

## Abstract

The perception of football as a male-dominated sport by society, coupled with the socio-cultural and economic barriers faced by women, has constrained their presence in the domain of football and revealed the manifestation of gender norms within the sport. This exclusion further masculinizes sport, negatively affecting social unity and cohesion, and deepening inequality within sport. Within this context, the study seeks to reveal how football fans perceive female and male referees through metaphorical representations. Participants, selected using purposive sampling, are individuals who regularly attend football matches and have experience watching games officiated by female football referees. The research employed a phenomenological approach to analyse metaphors generated by 352 football fans regarding female and male referees. Data were collected online through the Google Forms platform, which was accessible only to the researcher via password-protected access. During the analysis process, metaphors were coded, categorized, and transformed into meaningful interpretative formats. Results indicate that female referees are predominantly described with metaphors associated with sexist objectification, such as “flower”, “rose”, and “queen.” Female referees are represented by social roles and stereotypes metaphors like “mother,” and “gold,” yet they are also confronted with violence and disparaging metaphors such as “trash” and “chaos.” Conversely, male referees are perceived through metaphors evoking strength, toughness, and authority, including “lion”, “stone”, “authority”, “king”, and “leader.” These metaphorical representations highlight the persistence of gender norms within sport, demonstrating how women’s professional competencies are overshadowed by societal codes. Moreover, they are depicted as figures of power and discipline, reflecting masculinity within the sporting context. Ultimately, the research seeks to raise awareness about gender-based perceptions and foster transformation towards greater gender equality in sport.

## 1. Introduction

The presence of women in football has been shaped from past to present by male-dominant and gender-biased perceptions ([46]). Gender norms profoundly influence practices within sports and modes of participation, notably limiting the position of women in male-dominated sports such as football ([81]; [65]; [60]). Women’s participation in football faces numerous adversities, primarily socio-cultural, economic, and structural barriers ([28]). Women marginalized on the football field encounter sexist attitudes and face discrimination ([23]; [50]) and even physical threats ([72]). The monopolization of elements within football (players, coaches, referees, administrators, etc.) by men ([77]; [81]) has diminished the visibility of women in the field ([92]). Particularly, decision-making roles such as refereeing are not considered appropriate, especially for women, shows that social perceptions and institutional structures are built with sexist codes ([33]). In this context, the visibility of women in football and refereeing represents not only a sporting achievement but also a struggle for gender equality and the dismantling of prejudiced social identity taboos ([37]).

This identity struggle should be read not merely as an individual valuation but as an essential step towards transforming social representations embedded within sports culture ([52]). In the modern sociology of sport literature, social identity theory holds a central role, particularly in understanding power relations, the shaping of subjectivities, and the reproduction of social norms in sports within the context of gender ([11]; [30]). Social identity theory argues that individuals’ senses of belonging are shaped through group memberships ([86]). Its counterpart in sports sheds light on how fan, athlete, and referee identities are socially constructed ([11]; [9]). In this regard, female referees encounter a unique form of representation both as a marginalized group within the male-dominated structure of football and within the symbolic language of sports ([74], [73]). The experiences of female referees in football, a sport strongly associated with masculinity, provide essential examples related to positioning oneself within masculinity norms and the processes of social acceptance ([44]; [69]; [29]; [32]). Female referees are constantly compelled to negotiate their identities within the context of “men’s football,” which points to contradictions between gender norms and professional activities ([38]; [73]). The impact of these norms on women’s existence in sports is elaborated in deeper studies ([80]; [45]; [61]).

Within this theoretical framework, actions directed toward female referees can be examined under different categories. First, disparaging and exclusionary discourses are particularly noteworthy. Social judgments suggesting that women do not understand football or that they are solely associated with domestic roles directly undermine the professional competence of female referees ([38]). Another related category, sexual objectification, demonstrates that female referees are evaluated based on their bodies rather than their professional identities ([69]). In addition, practices of verbal violence and insult lead to challenges against the authority of female referees, while threats of physical violence or harassment emerge as serious pressures that undermine their presence in the sporting environment ([72]; [50]). These negative categories systematically hinder the processes of social acceptance for female referees. Ultimately, this theoretical framework contributes to understanding the experiences of female referees by examining how different gender identities are constructed in sport and how social perceptions and practices related to these identities can be analyzed within a category-based structure ([80]).

The media plays an intermediary role in the reproduction of exclusionary discourses ([76]; [97]). The use of sexist language in sports media, together with the visual representations of female athletes and referees, significantly shapes societal perceptions ([58]; [75]). The marginalization of female referees, or their direct exposure to sexist commentary, negatively influences their visibility in sport and the processes of social acceptance ([98]; [71]; [94]). From a critical perspective, however, gender studies and intersectionality approaches reveal that the experiences of female referees are shaped not only by gender but also by other identity dimensions such as sexual orientation and ethnicity ([66]; [15]; [20]). Accordingly, the sexist and homophobic pressures faced by female referees represent one of the visible manifestations of this intersectionality in sport ([50]; [38]). However, despite all these identity struggles and the professional barriers constructed, promising developments have emerged in the last decade regarding the recognition and support of female referees’ professional competencies.

FIFA and UEFA, as the main actors in football governance, have announced various policies aimed at ensuring gender equality in the recent period ([35]; [95]). One of these policies involves positive actions in favour of increasing the number of female referees. For instance, despite a 72% increase in the number of female referees in England between 2016 and 2020 due to updated policies, women are still underrepresented, especially in high-level men’s football matches ([90]). In 2017, India’s All India Football Federation (AIFF) launched its first academy dedicated to training female referees, offering education and elite training to support their development globally. Football Australia, in 2019, announced a ten-year Gender Equality Action Plan focusing on increasing female participation, leadership roles (such as referee), and improving facilities for women and girls. While international organizations and some countries have made efforts to promote gender equality in football refereeing, Türkiye has been a pioneer since much earlier. Unlike these recent developments, in Türkiye, women’s football refereeing emerged in the late 1960s under the pioneering leadership of Drahşan Arda, who holds the distinction of being the world’s first female football referee, officially recognized by FIFA, demonstrating the potential of women in this field ([24]). However, despite such an egalitarian start for Turkish football, gendered structural barriers continue to persist today. Female referees in Türkiye hardly find a place in the top leagues, and they struggle to exist with low representation rates in decision-making bodies and observer positions ([87]). In the “Women’s Football Strategy” report, which includes the Turkish Football Federation’s projections for the period 2024–2027, it is stated that there are currently 592 licensed female football referees in Türkiye. This figure corresponds to only about 10% compared to male referees according to the most recent data, reflecting a significant gender gap ([89]). These women officiate under challenging conditions and often face difficulties in receiving adequate support from both official institutions and civil society, as highlighted in recent academic studies examining the structural barriers and discrimination faced by female referees in Türkiye ([33]). According to FIFA’s 2023 Women’s Football Report, the number of female referees in Turkey ranks 7th among European countries and 13th worldwide ([36]). Despite this, at the highest level (Turkish Super League), no female referees are officiating, highlighting the ongoing gender disparity in top-tier football in Türkiye ([88]). All these challenges faced by women within football are linked to complex sociological and cultural dynamics that cannot merely be explained by physical capabilities ([31]). Gender norms narrow women’s roles in sports while creating social spaces where masculinity is reproduced through sports practice ([4]; [60]).

One of the areas contributing to the production of masculinity is undoubtedly language and discourse ([1]). While individuals attempt to express their feelings and thoughts through language, this can be hindered by context, situation, or time. In this respect, enabling individuals to reveal their subconscious thoughts and convey emotions and ideas through examples, analogies, or brief metaphoric expressions enriches communication ([63]). [91] ([91]) emphasize that metaphors are constantly used in everyday language and are much more than just a poetic device. They state that metaphors help us categorize the world and create cognitive frames about events. [56] ([56]) explain that a metaphor functions as both a rhetorical figure and a symbolic expression, linking abstract ideas to concrete images. By creating comparisons between different concepts, metaphors uncover hidden meanings and elicit deep psychological effects. [53] ([53]) further assert that metaphors play a decisive role in conceptual modeling and cognitive processes. For example, after a natural disaster, the metaphor of a “war zone” is frequently used in the media to describe the situation. In this regard, metaphors represent a significant element of discourse in the processes of understanding and conceptualizing the world, enhancing the perceived reality of individuals by stimulating creativity through analogical associations ([16]; [54]).

The field of sports is also one where metaphors are used in various ways ([39]). Metaphors, with their structure that reinforces the social and cultural dimension of sports ([42]), emerge as a significant tool for revealing the underlying reasons of gender equality issues. Thus, by interpreting the formation of perceptions towards female referees in the still male-dominant football culture through imaginations, this study aims to reveal how gender norms are constructed over the institution of refereeing.

Building on the existing literature, this study aims to explore, through metaphors, the feelings and thoughts of football fans who have experienced matches officiated by both female and male referees. The study reveals that perceptions of football refereeing are shaped not only by performance-based evaluations but also through metaphorical representations linked to gender roles. Moreover, by utilizing metaphors that reflect subconscious perceptions, emotions, and social norms, it enables a more nuanced and layered understanding of football fans’ attitudes toward referees. Thus, this approach offers a unique perspective and demonstrates the study’s potential to address a significant gap in the literature.

## 2. Method

### 2.1. Research Design

This study was designed based on the phenomenology approach, one of the qualitative research methods. The phenomenology design focuses on phenomena that are consciously experienced but lack in-depth understanding ([21]). Phenomenological research places the concept of ‘experience’ at its core with critical importance ([7]). According to [14] ([14]), data in such studies aim to reveal individuals lived experiences and the meanings they attach to these experiences. This research has a key aspect in that it emphasizes not only individuals’ personal experiences but also the commonalities among these experiences ([62]). In this regard, the study demonstrates how participants socially internalize the concept of the referee through metaphors. The metaphor analysis was conducted using Schmitt’s systematic approach for deriving ‘collective metaphorical models’. Our dataset consisted of metaphorical expressions that have semantic relationships with the term ‘referee’. In this way, underlying abstract conceptualizations are revealed through their concrete manifestations ([78]). Furthermore, participants were asked not only to generate metaphors but also to justify these metaphors using an open-ended method. According to [18] ([18]), qualitative studies using metaphor elicitation techniques offer the opportunity to uncover unconscious thoughts that reflect participants’ deepest thoughts and feelings about a concept. In this respect, it supports the exploratory nature of the phenomenological approach, which focuses on experiences. In addition to all this, the results are generally presented descriptively and supported by direct quotations. The researcher attempts to convey participants’ experiences impartially, without including personal preconceptions (bracketing) ([96]).

In the study, football fans were asked to create metaphors related to female and male referees and to justify these metaphors. Participants produced metaphors relevant to the given topic and expressed the bases of these metaphors in written form.

### 2.2. Research Participants

The population of the study consists of staff and students at Yozgat Bozok University, a state university in Türkiye. Criterion sampling, a purposive sampling technique, was preferred for sample selection. Criterion sampling is based on selecting individuals, events, objects, or situations that have specific characteristics related to the research problem ([70]). These participants are individuals who regularly attend football matches and have experience watching games officiated by female football referees. Accordingly, a total of 372 individuals who identified themselves as football fans and actively followed Türkiye’s Super League matches were selected for sample representation. Twenty missing or failed records were identified and eliminated, and the dataset consisted of 352 valid participants.

Among the participants, 176 (~46%) are female and 196 (~54%) are male. The majority of participants (*n* = 249, ~66%) belong to the 18–22 age group. Distribution according to the teams they support is as follows: Galatasaray (*n* = 151, ~40%), Fenerbahce (*n* = 111, ~29%), Besiktas (*n* = 78, 21%), Trabzonspor (*n* = 7, ~1%), and other Turkish Super League teams (*n* = 25, ~6%).

### 2.3. Data Collection

Data collection tools were integrated into the Google Forms system, which can only be accessed by the researcher with a password. The prepared form consisted of three parts. The first page of the online form contained participant information sheet and a participant agreement form. For participants who did not digitally approve the agreement form or did not give consent for this research, the survey was automatically terminated. The second part requested fans to create metaphors related to female and male referees and to justify these metaphors. Participants were asked to complete the expressions “A female referee is like … because …” and “A male referee is like … because …” in a total of four questions. The term “like” was used to express the metaphors chosen by the participants, while “because” aimed to elicit justifications for these metaphors. The third and final section included demographic questions related to gender, age, and the football team they support. The survey form did not include any questions other than socio-demographic information and questions related to the data collection tools, and no personal data were collected. To ensure credibility, participants were not directed in any way ([57]). Data were collected between 25 March 2025 and 1 May 2025. To protect participant honesty, no information that might reveal identities was included in the collection period, and it was explicitly stated that data access would be restricted to researchers only.

### 2.4. Data Analysis

The data analysis in this study followed a systematic qualitative approach aligned with phenomenological principles. Specifically, the metaphor analysis was used as a phenomenological lens to explore the participants’ lived experiences and perceptions ([8]; [13]). Also, trustworthiness in qualitative research is a fundamental criterion ([57]), and various methods recommended in the literature were employed to ensure validity and reliability in qualitative design ([5]). In this regard, the data analysis process involved defining metaphors, systematically categorizing them to form thematic clusters, coding by multiple experts (peer reviewers), and finally, the inclusion of direct quotations from participants to illustrate and substantiate the themes. Frequency analysis was carried out to determine the frequency of metaphors and to objectively support their distribution across categories. This approach supports methodological triangulation, increasing the rigor and validity of the study ([25]). The process was conducted rigorously and sequentially to ensure confirmability.

Firstly, the data were transferred from Google Forms database to Microsoft Excel. The data were included in the analysis process without any modification to ensure credibility ([93]). The metaphors were subjected to open coding by the researcher, based on the meaning of each metaphor, and were classified under specific categories by grouping those with semantic coherence using the axial coding technique ([85]). Then, the dataset was sent to independent experts experienced in qualitative research who evaluated the metaphors without prior knowledge of the researcher’s classifications. Finally, the coding by the researcher and the expert were compared, and the inter-coder reliability was assessed through Kappa analysis. The obtained Kappa coefficient was 0.80, indicating a good level of agreement between coders ([17]). The review of data by different experts increased trustworthiness and dependability of the study. Finally, to ensure credibility, three exemplary statements were provided for each category, and participants’ statements were included in the report without any modification. In interpreting the data, the created metaphors and the categories they were included in were examined.

Researchers’ sports experience and their knowledge of metaphor analysis contributed to the credibility of the study through reflexivity. In addition, periodic short-interval meetings and peer review sessions further supported dependability and confirmability.

### 2.5. Language Considerations

The study was carried out in Turkish, the native language of all participants. Therefore, no language barriers or comprehension difficulties related to the use of a second language were present. However, as the study was published in English, linguistic support was provided by an expert linguist in the field to ensure that participants’ thoughts and experiences were accurately conveyed without distortions arising from translation. According to [84] ([84]), this approach minimizes the risk of language affecting the validity of the data.

### 2.6. Ethical Statement

Prior to data collection, an application was submitted to the Social and Human Sciences Ethics Committee Yozgat Bozok University, and ethical approval was obtained with decision number 23/31 dated 19 March 2025.

## 3. Results

In this section, the categories generated from the analysis, the most frequently used metaphors, and the reasons for their production are presented in detailed tables. In Table 1, the categories based on the most frequently used metaphors by the research participants are given.

Table 2 presents the metaphors associated with female referees and details the reasons justifying their use by football fans.

In the study, female referees were most frequently described with metaphors such as ‘flower’, ‘rose’, and ‘queen’, symbolizing qualities of delicacy, elegance, and fragility, which serve to metaphorically characterize them as sensitive and delicate, reinforcing gendered stereotypes of femininity. This indicates that even in professional contexts where evaluations should be gender-neutral, women are still associated with these types of traits. These associations highlight the strong influence of gender roles in fields like sports that should ideally be gender-independent. [82]’s ([82]) research on metaphor use in sports commentary reveals that sports language and narratives reproduce women as aesthetic and fragile figures, while [71]’s ([71]) study on sports media shows that women are depicted with sexist narratives emphasizing naivety and fragility. Additionally, [48] ([48]) found that female athletes on social media and in sports news are represented with a “strong but graceful” image. These findings align with our study’s results, suggesting that societal representations of female referees are grounded more in gender roles than professional competency. Some metaphors are observed to reinforce biased and gender-discriminatory patterns. It can be argued that, besides explicitly sexist and derogatory metaphors, even those framed in seemingly positive ways ultimately function to preserve and legitimize structures that reproduce gender inequality and sustain male dominance.

The professional role and management category includes metaphors emphasizing female referees’ leadership qualities, such as “power” and “manager.” [6]’s ([6]) research found women to be more successful than men in decision-making stages, with men showing hesitation and avoidance. [29] ([29]) also state that female referees take on strong leadership roles, supporting the metaphors found in our study.

The findings also revealed that referees are socially evaluated by separating them into women and men. While male referees are not metaphorically associated with fatherhood, female referees are likened to “mother” and “gold,” symbolizing creators of respect and love in the sports environment. This corresponds with women being defined by social and emotional roles such as emotional, embracing, and valuable. [79] ([79]) similarly found that women in sports are often identified with nurturing roles like sister or mother through social dynamics. At this point, social identity theory notes that such othering may weaken in-group solidarity and cause crises in female referees’ professional identities ([11]). Additionally, sports narratives are also shaped by sexist codes that confine femininity to certain stereotypes ([46]; [10]). Media fuels these negative cycles ([76]; [97]). [64] ([64]) emphasize the explicit highlighting of femininity within sports media, and [12] ([12]) stress that discourses are produced under male dominance with masculinity and violence themes. [29] ([29]) state the meaningless nature of distinguishing referees by gender and affirm the gender-neutrality of refereeing roles. Correspondingly, the International Olympic Committee (IOC) in its depiction guide (Gender-equal, Fair And Inclusive Representation In Sport) stresses that terms defining or emphasizing gender are divisive and offers guidelines for appropriate use ([49]). The Association for Women in Sport and Physical Activity (KASFAD) also advocates avoiding the perception of sports, organizations, and professions as “male domains” and recommends gender-sensitive language use only when necessary ([51]). Considering all these challenges, there is strong consensus in gender identity studies on the need for policies to overcome obstacles faced by female referees ([55]; [22]; [43]). Sports organizations developing concrete support and visibility strategies against stereotypes related to female professional identities will accelerate this socio-structural transformation ([77]).

In the violence and disparaging speech category, participants who believe that women should not participate in male-dominated fields like football and that women’s place is at home explain this viewpoint using metaphors such as “trash” and “chaos.” Supporting this finding, research addressing sexist perspectives does exist. For instance, [38] ([38]) report that female referees often face sexist discourses like “Women cannot referee!.” Similarly, [69] ([69]) highlights derogatory expressions such as “Go back to the kitchen!” directed at female referees, which threaten their professional identities. In contrast to various discourses that weaken women’s social positions, such discrimination can also be applied to men. For example, [27]’s ([27]) study on sports media notes that male referees who make mistakes are mocked using feminine-associated terms like “eksik etek(…skirt)” (a term with a sexist pejorative meaning in Turkish slang) indicating that feminine traits are used to demean men as well. These findings parallel the expressions forming the “violence and disparaging speech” category.

Table 3 includes the metaphors associated with male referees and the reasons justifying their use.

In the study, metaphors related to male referees revealed that they are generally associated with traits such as power, toughness, justice, leadership, and professional authority. Metaphors like “stone” or “power” reference their physical and mental resilience. These metaphors reinforce social perceptions of men being “naturally” and “rightfully” placed in roles involving physical strength, leadership, and discipline. In this sense, men are also assigned a social identity and responsibility that may not always stem from their own will, but rather from gendered expectations imposed by society. [83] ([83]) describes sports as an entity intertwined with masculinity and power. [1] ([1]) support these findings by showing how men reproduce social dominance through sports narratives that equate masculinity with physical strength, toughness, and leadership. This is a clear example from the perspective of social identity theory of masculinity norms being repeatedly produced in sport ([41]; [19]; [59]).

In terms of professional role and management, male referees are directly linked to decision-making positions (referee, representative, manager) and are portrayed as authoritative figures who enforce discipline, reflecting their acceptance as part of the main normative group in the sports context. [68] ([68]) found that leadership and decision-making dominance remain with men in sports leadership, supporting the current study’s results. However, many issues can be overcome with technological advancements. For example, the Video Assistant Referee (VAR) system, implemented by FIFA in 2018, aims to prevent erroneous decisions and possible subjective approaches by referees during matches ([47]). Despite this, [81] ([81]) found that such technological solutions have not completely eradicated sexist stereotypes in societal identities of female referees.

Regarding social roles and stereotypes, male referees were regarded as individuals who guide on the field, demonstrating high leadership skills and charisma. [34] ([34]) explain men’s representation as symbols of leadership and authority within a conceptual framework. While these attributes refer to leadership skills and charisma, they can sometimes be destructive and characterized by harshness. This also corresponds to the category classified as violence and rude speech. At that category, male referees were associated with negative metaphors such as “dictator” “rude” “egoistic” and “favouritism” indicating that some participants perceive male referees as biased or harsh. In some studies, in the field of management, it has been shown that charismatic leadership, when taken to excess, can be abused and lead to destructive decisions and ethical problems ([67]; [40]; [2]). [26] ([26]) note that male referees are often viewed as harsh and aggressive, sometimes even biased, which aligns with the current study’s findings.

Both genders shared metaphors in the categories of “justice and impartiality” and “other analogies,” indicating some common metaphorical and contextual consensuses. Metaphors in the “justice and impartiality” category, such as “scale” and “equality”, used for female referees, suggest their more objective and ethical behavior. However, [3] ([3]) argues, based on match reports published in The Guardian, that metaphors such as “justice” and “scale” used to describe female referees’ impartiality and honesty fundamentally reflect a gender-biased perspective. This shows how gender norms restructure even the concept of justice according to gender. For male referees, the corresponding category consists of authoritarian metaphors such as “punishment” and “discipline,” encouraging their positioning as enforcers of sanction and power in social identity contexts. In the “other analogies” category, for female referees, metaphors such as “lion” and “eagle”, were preferred within the context to highlight possessive and keen observational traits, showing a shared use of common metaphors among the participants. Similarly, male referees were described with metaphors such as “lion”, “fox”, and “robot” emphasizing strength, cunning, and mechanical execution of duties. From this perspective, it can be argued that the gender-based differentiation in professional roles is diminishing, and social identities are moving away from discrimination towards a more egalitarian viewpoint. [99] ([99]), in their study comparing the match performances of female and male referees, state that there were no significant differences between female and male referees in managing objective variables such as goal decisions, fouls, penalties, offsides, and card usage. These results reveal that female referees are neither superior nor weaker than male referees in terms of physical or decision-making performance, thereby limiting the impact of gender-based discrimination on professional roles. This has also been observed previously within the category of professional roles and management. In this category, the metaphor ‘manager’ was commonly used, emphasizing the professionalism of referees rather than a gender-biased distinction.

## 4. Conclusions and Recommendations

This study addresses societal gender perceptions of female and male football referees within the framework of social identity theory. This theoretical framework guides understanding how metaphors directed at referees in sports are structured according to gender norms and how secondary identities are shaped accordingly.

At first glance, metaphors attributed to female referees may appear complimentary; however, they fundamentally reinforce sexist discourse by objectifying women and confining them to traditional feminine values. On the other hand, metaphors associated with male referees reflect cultural masculinity ideals. Nevertheless, these representations also impose restrictive expectations on men to always be strong, resilient, and authoritative. Therefore, the metaphors related to both female and male referees contribute to the perpetuation of normative gender roles that limit individual identity and professional autonomy. Recognizing this dual impact is crucial for advancing gender equality in sport, as it requires addressing not only how women are marginalized but also how dominant masculinity norms affect men.

From the perspective of social identity theory, individuals’ identities are shaped not only by their group memberships but also by the secondary identities within those groups. In sport, gender is a complex social construct influenced by these layered identities. The metaphors attributed to female and male referees do more than describe they shape self-perception and social roles, reinforcing traditional gender norms. These identity constraints limit personal expression and maintain existing power structures, reproducing inequalities in sport. Achieving gender equality requires challenging these identity frameworks to transform both individual perceptions and broader social norms in sport. Thus, social identity theory provides a crucial lens for understanding how deeper cultural and structural factors influence identity in sports.

In light of these findings, the following concrete steps are recommended to transform gender-based perceptions toward female and male referees:Football federations’ efforts to increase the visibility of female referees and emphasize the gender neutrality of the profession should be encouraged through collaborations within their organizations and with the media.Action plans aimed at increasing the representation of female referees in top-tier leagues and decision-making bodies such as the Referee Board should be developed and implemented effectively.Sports federations should establish gender-sensitive portrayal guidelines for media, as the IOC did in 2024, and encourage compliance among stakeholders (club officials, commentators, broadcasters, etc.) to promote equality in language and representation.

These recommendations hold significant importance not only within the scope of football but also across all sports federations and at the societal level, aiming to foster gender equality and reinforce the professional identities of women.

## Figures and Tables

**Table 1 behavsci-15-01359-t001:** Metaphor-based categories related by gender.

Female Referees	Male Referees
Sexist Objectification Justice and Impartiality Professional Role and Management Social Roles and Stereotypes	Masculine Objectification Justice and Impartiality Professional Role and Management Social Roles and Stereotypes
Violence and Disparaging Speech	Violence and Rude Speech
Other Analogies	Other Analogies

**Table 2 behavsci-15-01359-t002:** Metaphors related to female referees and their justifications.

Category	Metaphors	Frequency	Justification (Because…)
Sexist Objectification	Flower	63	…someone who exists in a particular atmosphere and makes that atmosphere unique. …she is both a leader and does not lose her femininity. …female referees are delicate and gentle.
	Rose	22	
	Beautiful	4	
	Queen	3	
Justice and Impartiality	Justice	24	…they apply in-game ethical rules more objectively. …women are just people with clearer boundaries. …they make fair and impartial decisions.
	Scale	9	
	Equality	4	
Professional Role and Management	Power	3	…women are always those who can control the chaos within football. …they are strong, brave, and possessive. …act according to a set of rules.
	Manager	3	
Social Roles and Stereotypes	Mother	6	…I believe matches led by female referees will be more effective and less abusive, because everyone has a woman like a mother or sister in their family. …female referees are so valuable that they create an environment of respect and love, thereby increasing youths’ interest in sports. …they are rare, valuable, and precious.
	Gold	6	
Violence and Disparaging Speech	Trash	6	…there is no place for female referees in this industrial football. …they cannot manage the game with the same perspective as men. …women should stay in their kitchens.
	Chaos	6	
Other Analogies	Lion	10	…they are possessive, strong, and noble. …they do a very courageous job in a male-dominated environment. …they observe well during the matches.
	Eagle	3	

**Table 3 behavsci-15-01359-t003:** Metaphors related to male referees and their justifications.

Category	Metaphors	Frequency	Justification (Because…)
Masculine Objectification	Stone	10	…knows how to get up after being hit. …are ruthless but just individuals. …men are physically strong.
	Tough	9	
	Authority	9	
	Power	7	
Justice and Impartiality	Justice	17	…since men are more committed to football, they find it easier to have problems with impartiality. …punish when necessary. …maintain order and ensure rules are followed.
	Equality	4	
	Fair	3	
Professional Role and Management	Referee	20	…symbol of football is male. …male referees are authorities who enforce the rules and maintain discipline more than females. …their stance is serious and professional.
	Representative	7	
	Manager	4	
Social Roles and Stereotypes	King	4	…is a guide on the field. …better than women at leadership and decision-making. …by nature, men are leaders.
	Leader	3	
	Charismatic	3	
Violence and Rude Speech	Wood	8	…since childhood, they support their fathers’ teams, so they tend to favor them in matches. …work like a guillotine, cutting and throwing away. …not always fair because they decide based on fan pressure and match conditions.
	Dictator	6	
	Enemy	4	
	Rude	4	
	Egoistic	3	
	Favoritism	3	
Other Analogies	Lion	22	…strong and determined. …unsurpassed in cunning. …moves like a machine.
	Animal	5	
	Fox	5	
	Robot	5	

## Data Availability

The raw data presented in the study are openly available in https://osf.io/yxtfj/?view_only=3dc6de5d596d431fbcd98a4462c9d329. (accessed on 25 August 2025).

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
