# Peer review of "The Field of Gender Through Metaphors: The Dilemma of Female and Male Referees in the Minds of Football Fans"

_behavsci, 2025, doi:10.3390/bs15101359_

Round 1

Reviewer 1 Report

Comments and Suggestions for Authors

Thank you for giving me the opportunity to review this article. I found it interesting and I'm sending some comments that I hope will help the author.

Pages 2-3 - Section 2

Perhaps it would be interesting to raise and discuss within a theoretical framework possible categories of concepts/perspectives toward women in sports: contempt (women don't know about soccer; women are only good for housework); sexual objectification; verbal violence, insults; threats of physical violence or rape... and observe whether these issues are represented in the discourse or whether other, more positive options emerge, such as support or vindication. In this way, the discussion can confront traditional discourses with the results found in the research.

Page 3 - lines 124 and following

Does the author think it would be interesting to present the concept of metaphor and the various possibilities that can be found in research analyses in more detail?

Page 3 - lines 135 and following

We understand that all participants have lived experiences regarding the situation of women in refereeing? Would it be appropriate to clarify both the subject of the study and the selection of participants and their relationship with football? I would appreciate some explanations, as I find this part complicated/confusing.

Pages 5-6

I think the variables in the discourse classification can be adjusted. For example, "they make fair and impartial decisions" appears under "professional role" and would probably be more appropriate under "Justice and Impartiality."

I find the variable "Elegance and Aesthetics" highly questionable, when in reality it can be interpreted as a reification of women and a sexist disdain/prejudice.

I find it questionable to include in a section like "Value and Worth" values ​​that, from a supposedly positive perspective, reiterate the most basic stereotypes of women as mothers.

I find it questionable to consider offensive, contemptuous, and violent discourses "negative attributes."

I find it questionable that "...they are strong, brave, and possessive" is classified as "Professional Role and Management" and "...they are possessive, strong, and noble" as other analogies.

Therefore, I believe the classification should be thoroughly reviewed.

Curiously, the development of the discussion confirms these negative, stereotypical views, loaded with prejudice and violence to maintain the status quo that does not correspond to the denomination of the different classifications.

Page 7

According to the discussion "male referees who make mistakes are mocked using feminine-associated terms like "... skirt," indicating that feminine traits are used to demean men as well" I would consider it more appropriate to raise the "violence" that it entails rather than "negative attributes."

I believe the discussion and results section includes the entire theoretical foundation (representation of male and female referees) in social discourse.

For example, from page 8, line 256 to page 9, line 303, the different classifications are supported by the bibliography. However, I believe it would be more appropriate to work this foundation into the theoretical framework, and in the results and discussion section, compare the results with the most appropriate bibliographic references presented previously.

page 9

I think the conclusions would be more powerful if we could develop and argue the comment "The presence of female referees in the male-dominated field of football leads not only to questioning their professional skills but also to the reproduction of societal gender norms."

page 9

I am surprised that the conclusions give a fundamental role to the media, when I find no references or discussion about this fundamental role throughout the article.

Author Response

Comments 1: Pages 2-3 - Section 2; Perhaps it would be interesting to raise and discuss within a theoretical framework possible categories of concepts/perspectives toward women in sports: contempt (women don't know about soccer; women are only good for housework); sexual objectification; verbal violence, insults; threats of physical violence or rape... and observe whether these issues are represented in the discourse or whether other, more positive options emerge, such as support or vindication. In this way, the discussion can confront traditional discourses with the results found in the research.

Response 1: Thank you for pointing this out. In response to your feedback, the information on pages 2 and 3 has been revised, the theoretical framework has been positioned in the appropriate sections, and, as per your guidance, the negative practices directed toward women have been categorized and presented under specific headings. In addition, organizational practices that aim to empower women have also been included.

Comments 2: Page 3 - lines 124 and following; Does the author think it would be interesting to present the concept of metaphor and the various possibilities that can be found in research analyses in more detail?

Response 2: Thank you for addressing this point. In the study, the concept of metaphor has been expanded between lines 138 and 171 on page 3, where detailed information about its applications and purposes has been provided. Additionally, comprehensive explanations regarding the function of metaphors have been included.

Comments 3: Page 3 - lines 135 and following; We understand that all participants have lived experiences regarding the situation of women in refereeing? Would it be appropriate to clarify both the subject of the study and the selection of participants and their relationship with football? I would appreciate some explanations, as I find this part complicated/confusing.

Response 3: Thank you for your contribution. The details regarding the selection of participants have been expanded in the research design section on page 4. Additionally, on page 5, line 204, a sentence has been added that more clearly defines the characteristics of the participants.

Comments 4: Pages 5-6; I think the variables in the discourse classification can be adjusted. For example, "they make fair and impartial decisions" appears under "professional role" and would probably be more appropriate under "Justice and Impartiality."

Response 4: Thank you for your correction. The relevant quotation has been moved under the appropriate category and its adjustment has been made. It has been placed in the suitable position under the categories Table 2 on page 7.

Comments 5: I find the variable "Elegance and Aesthetics" highly questionable, when in reality it can be interpreted as a reification of women and a sexist disdain/prejudice.
Response 5: Thank you very much for this valuable suggestion, which has strengthened my study. Following your guidance, the categories related to female and male referees have been reviewed and revised. In line with your suggestion, the "Elegance and Aesthetics" category has been strengthened as "Sexist Objectification." These revisions can be seen in Table 2 on page 7 and Table 3 on page 9.

Comments 6: I find it questionable to include in a section like "Value and Worth" values ​​that, from a supposedly positive perspective, reiterate the most basic stereotypes of women as mothers.
Response 6: Thank you for your suggestion. In line with your valid critique, the "Value and Worth" category has been updated to "Social Roles and Stereotypes," and we believe this revised category is stronger. You can see this change in Table 2 on page 7.

Comments 7: I find it questionable to consider offensive, contemptuous, and violent discourses "negative attributes."
Response 7: Thank you for your critique, which has strengthened my work. In line with your suggestion, the "negative attributes" category has been updated to "Violence and Disparaging Speech" to create a stronger emphasis. You can find this update in Table 2 on page 7.

Comments 8: I find it questionable that "...they are strong, brave, and possessive" is classified as "Professional Role and Management" and "...they are possessive, strong, and noble" as other analogies.
Response 8: Thank you for expressing your valid concern. However, the reason for creating the "Other Analogies" category is to distinguish metaphors that differ in their literal meaning from the other categories. Although participant views under this heading may share similarities with other sections in the justification, they are primarily based on animal metaphors, which is why they have been classified separately. Additionally, the metaphors in this category mainly reflect the referees' perception of their professional skills as strong and are evaluated from a positive perspective. For these reasons, this category was established.Thank you for expressing your valid concern. However, the reason for creating the "Other Analogies" category is to distinguish metaphors that differ in their literal meaning from the other categories. Although participant views under this heading may share similarities with other sections in the justification, they are primarily based on animal metaphors, which is why they have been classified separately. Additionally, the metaphors in this category mainly reflect the referees' perception of their professional skills as strong and are evaluated from a positive perspective. For these reasons, this category was established.

Comments 9: Curiously, the development of the discussion confirms these negative, stereotypical views, loaded with prejudice and violence to maintain the status quo that does not correspond to the denomination of the different classifications.
Response 9: Thank you for your critique. The perception you mentioned may have arisen because most studies in the discussion section are presented in this manner; however, we have also tried to emphasize, as much as possible, how strongly and successfully women have established their identities in strategic positions. Additionally, on page 8, line 308, the studies by Atılgan and Tükel highlight that women have been shown to be more successful than men in decision-making positions. Furthermore, following your guidance, strengthening additions have been made on page 8.

Comments 10: Page 7; According to the discussion "male referees who make mistakes are mocked using feminine-associated terms like "... skirt," indicating that feminine traits are used to demean men as well" I would consider it more appropriate to raise the "violence" that it entails rather than "negative attributes."

Response 10: Thank you for your suggestion for improvement. Following your guidance, the expression "...skirt" was reverted to its Turkish version since it does not have an exact English equivalent and has been evaluated under the "violence and disparaging speech" category, as you can see on page 9, line 350.

Comments 11: I believe the discussion and results section includes the entire theoretical foundation (representation of male and female referees) in social discourse. For example, from page 8, line 256 to page 9, line 303, the different classifications are supported by the bibliography. However, I believe it would be more appropriate to work this foundation into the theoretical framework, and in the results and discussion section, compare the results with the most appropriate bibliographic references presented previously.

Response 11: I find your critique at this point very meaningful. The research has been theoretically simplified, and the discussion and results sections have been supported through social identity theory. In particular, expressions emphasizing physical and mental endurance, such as lion, stone, and mountain, refer to gender-based roles specific to male referees; the highlighting of masculine traits, and perceptions like physical strength and leadership as characteristics based on male identity, are explained exactly by social identity theory. This theoretical basis has been discussed with references to GiazitsoÄŸlu (2024), Connell & Messerschmidt (2005), and Messner 1990.

Comments 12: page 9; I think the conclusions would be more powerful if we could develop and argue the comment "The presence of female referees in the male-dominated field of football leads not only to questioning their professional skills but also to the reproduction of societal gender norms."

Response 12: Thank you for your meaningful contribution. Following your guidance, revisions have been made under the conclusion and recommendations section on page 11, and the conclusion has been strengthened by emphasizing both identity theory and gender norms.

Comments 13: page 9; I am surprised that the conclusions give a fundamental role to the media, when I find no references or discussion about this fundamental role throughout the article.
Response 13: Thank you for your valuable contribution. In the recommendations section, a detailed explanation has been added between lines 86 and 101 on page 2 about how the media plays an important role in producing and constructing these social identities and gender norms.

Reviewer 2 Report

Comments and Suggestions for Authors

Abstract – more detail about data collection would be helpful. Context of the societal background would be helpful.

Introduction – the Messner paper 2002 is very outdated and there has been a lot written since – this is cited a number of times. FA stats were used but then the context was Turkey – societal contexts are essential to be discussed with gendered differences and might need to be clearer here. There is quite a jump from language to metaphors – I think you need more justification for the use of metaphors.

Literature – I am not sure you need the first paragraph, it doesn’t add to the argument. This section is not as strong at the introduction and is a little confused in terms of theory. You mention a number of theories which makes this section quite muddled and lacks depth. It might be more effective to just focus on one theory you are acknowledging. Perhaps you could remove this section and build on the introduction which flows and is well written and has depth and detail.

Somewhere either in the introduction or literature you need to acknowledge the cultural epoch in Turkey broadly and within football.

Method – Some consideration over language needed here – was it conducted in English? Could this have impacted the results if participants first language was not English? Unclear why validity is discussed within a qualitative study – credibility etc. would be more effective. Data analysis needed more depth how did you relate this to phenomenology? What system/model/type of analysis was used? Again terminology of validity and reliability needs to be revisited here. Unclear how statistics were used within this research design which appears to be qualitative. Unclear how data storage was ethical given it was done on google forms rather than a closed system. Phenomenological design does not seem to match the analysis and the results etc.

Results – from the methods it is unclear how the results were coded. Again I would reduce the number of theories here to make it more coherent and developed. I think the discussion could be narrowed to focus on one theory and get more depth of the analysis. There are very dated references within this section which could be much more updated. Some paragraphs are very short and could be developed further. I am not sure how much of a phenomenological design comes through with the results again, it seems add odds with what is presented. It feels more like an interpretivist case study from the methods and the results.

Conclusion – in the conclusion you mention two theories but you have also considered hegemonic masculinity. The recommendations are clear but the conclusion is very concise.

Author Response

Comments 1: Abstract – more detail about data collection would be helpful. Context of the societal background would be helpful.

Response 1: Thank you for pointing this out. Additional details regarding the data collection process have been incorporated into the abstract, along with an emphasis on the study’s aim to address the societal background issues.

Comments 2: Introduction – the Messner paper 2002 is very outdated and there has been a lot written since – this is cited a number of times. FA stats were used but then the context was Turkey – societal contexts are essential to be discussed with gendered differences and might need to be clearer here. There is quite a jump from language to metaphors – I think you need more justification for the use of metaphors.

Response 2: Thank you for these valuable comments about the Introduction section. 
1. Messner (2002) remains a foundational source in this field; however, the study also integrates recent references to provide a comprehensive contextualization, including Sipilä Kalte (2025), Hall et al. (2024), Skirbekk (2024), Hietala & Archibald (2021), Anderson & Hargreaves (2016), and Nordstrom et al. (2016). I consider it essential to acknowledge seminal works, even if somewhat dated, as they form the academic groundwork and honour scholarly heritage. Therefore, the arguments presented are not solely reliant on Messner (2002), and I kindly request that this be taken into account during your evaluation.
2. Reference was made to TheFA (2020) statistics not to present the general global context of female refereeing, but rather to illustrate that even in countries considered role models for gender equality and human rights, such as England (the so-called home of football), the representation of female referees in top-tier leagues still remains limited. In addition, examples from developments in diverse geographies and cultures such as Australia and India were also presented. Following this, the historical development and current situation of women referees in Turkey were examined. In this sense, the use of FA statistics aims to provide a comparative perspective between these countries and the case of Turkey, rather than to explain the overall global context. However, I fully acknowledge your point, and the section on the societal context in Turkey has been further elaborated in line with your suggestion.
3. The framework regarding the current status of female referees in Turkey has been expanded by utilizing football federation reports and academic sources. It has been particularly demonstrated through official source reports that female referees are unable to officiate in top-tier leagues in the Turkish football system.

Comments 3: Literature – I am not sure you need the first paragraph, it doesn’t add to the argument. This section is not as strong at the introduction and is a little confused in terms of theory. You mention a number of theories which makes this section quite muddled and lacks depth. It might be more effective to just focus on one theory you are acknowledging. Perhaps you could remove this section and build on the introduction which flows and is well written and has depth and detail. Somewhere either in the introduction or literature you need to acknowledge the cultural epoch in Turkey broadly and within football.

Response 3: 

1- Thank you for your contribution to strengthening my research. Following your suggestion, a single theoretical framework was determined, and the study was restructured based on social identity theory, providing a stronger theoretical perspective. Additionally, in line with your recommendations, the literature section was integrated into appropriate parts of the introduction, enhancing the readability of the study and creating a stronger introduction structure.

2- The current situation in Turkey, developments specific to football, and statistics were expanded and strengthened between lines 118-132 on page 3.

Comments 4: Method – Some consideration over language needed here – was it conducted in English? Could this have impacted the results if participants first language was not English? Unclear why validity is discussed within a qualitative study – credibility etc. would be more effective. Data analysis needed more depth how did you relate this to phenomenology? What system/model/type of analysis was used? Again terminology of validity and reliability needs to be revisited here. Unclear how statistics were used within this research design which appears to be qualitative. Unclear how data storage was ethical given it was done on google forms rather than a closed system. Phenomenological design does not seem to match the analysis and the results etc.

Response 4: I appreciate your valid criticisms regarding the method section and would like to inform you that I have thoroughly revisited this part to strengthen my study.

  1. To address concerns regarding the language of the research, I have added a section titled "Language Considerations" on page 6, line 267, where I provide detailed explanations.

  2. I have revisited and updated the terminology concerning validity and reliability in qualitative research. These updates have been further elaborated and reinforced under the headings of Research Design and Data Analyses.

  3. Within the Research Design section, I have included detailed information about the phenomenological approach. I clarified the connections, elaborated on the design details, and explained the rationale behind choosing this particular design.

  4. I have provided a detailed explanation regarding the rationale for the descriptive statistics used, specifically on page 6, line 243.

  5. To address ethical and procedural concerns related to the use of Google Forms, I have expanded upon these issues under the Data Collections section, including information about data access and management.

Comments 5: Results – from the methods it is unclear how the results were coded. Again I would reduce the number of theories here to make it more coherent and developed. I think the discussion could be narrowed to focus on one theory and get more depth of the analysis. There are very dated references within this section which could be much more updated. Some paragraphs are very short and could be developed further. I am not sure how much of a phenomenological design comes through with the results again, it seems add odds with what is presented. It feels more like an interpretivist case study from the methods and the results.

Response 5: Thank you for your valuable suggestions.

  1. Detailed information regarding the coding process has been provided between lines 240 and 258 in the methods section.

  2. Based on your recommendation, the results section has been updated to focus on a single theory, social identity theory, with the addition of current references.

  3. Details about the research design and the factors influencing its selection have been included under the Research Design heading in the methods section.

Comments 6: Conclusion – in the conclusion you mention two theories but you have also considered hegemonic masculinity. The recommendations are clear but the conclusion is very concise.

Response 6: Thank you for your meaningful contribution. Following your guidance, revisions have been made under the conclusion and recommendations section on page 11, and the conclusion has been strengthened by emphasizing identity theory and gender norms.

Round 2

Reviewer 2 Report

Comments and Suggestions for Authors

The revisions have developed the paper significantly.